# Strengthening Tuberculosis Services for Children and Adolescents in Low Endemic Settings

**DOI:** 10.3390/pathogens11020158

**Published:** 2022-01-26

**Authors:** Jeffrey R. Starke, Connie Erkens, Nicole Ritz, Ian Kitai

**Affiliations:** 1Department of Pediatrics, Division of Infectious Diseases, Baylor College of Medicine, Houston, TX 77030, USA; 2KNCV Tuberculosis Foundation, 2516 AB The Hague, The Netherlands; connie.erkens@kncvtbc.org; 3Department of Paediatrics and Paediatric Infectious Diseases, Children’s Hospital, Lucerne Cantonal Hospital, 6000 Lucerne, Switzerland; nicole.ritz@unibas.ch; 4Mycobacterial and Migrant Health Research Group, Department of Clinical Research, University of Basel Children’s Hospital, University of Basel, 4031 Basel, Switzerland; 5Department of Pediatrics, Division of Infectious Diseases, Hospital for Sick Children, University of Toronto, Toronto, ON M5G 1X8, Canada; ian.kitai@sickkids.ca

**Keywords:** tuberculosis, child, adolescent, tuberculosis services, low tuberculosis endemic

## Abstract

In low tuberculosis-burden countries, children and adolescents with the highest incidence of tuberculosis (TB) infection or disease are usually those who have immigrated from high-burden countries. It is, therefore, essential that low-burden countries provide healthcare services to immigrant and refugee families, to assure that their children can receive proper testing, evaluation, and treatment for TB. Active case-finding through contact tracing is a critical element of TB prevention in children and in finding TB disease at an early, easily treated stage. Passive case-finding by evaluating an ill child is often delayed, as other, more common infections and conditions are suspected initially. While high-quality laboratory services to detect *Mycobacterium tuberculosis* are generally available, they are often underutilized in the diagnosis of childhood TB, further delaying diagnosis in some cases. Performing research on TB disease is difficult because of the low number of cases that are spread over many locales, but critical research on the evaluation and treatment of TB infection has been an important legacy of low-burden countries. The continued education of medical providers and the involvement of educational, professional, and non-governmental organizations is a key element of maintaining awareness of the presence of TB. This article provides the perspective from North America and Western Europe but is relevant to many low-endemic settings. TB in children and adolescents will persist in low-burden countries as long as it persists throughout the rest of the world, and these wealthy countries must increase their financial commitment to end TB everywhere.

## 1. Introduction

Tuberculosis (TB) disease has become relatively rare in affluent countries and has retreated into specific population groups, especially foreign-born adults from high-burden countries and their offspring and indigenous people. It has largely become a disease of poverty and disadvantage. However, despite the availability of the most advanced technology, diagnosis is often delayed because of a lack of recognition and experience with the disease by practitioners. This leads to a detection gap, both in resource-rich and resource-limited countries.

Studying TB in low-burden countries can provide valuable information about the nature and transmission of the disease. Because of low incidence rates, the “background noise” is reduced, and the patterns of transmission can be seen plainly via contact tracing [1]. Reverse contact tracing often identifies the source case for a child. Less common modes and locations of transmission have also been uncovered by this strategy. Importantly, the infrequent transmission of *Mycobacterium tuberculosis* to others from young children with pulmonary TB has been verified [2,3]. When TB rates were higher, universal testing of children was often performed to find and treat TB infections before disease develops. As case rates declined, universal testing was replaced in many countries with selective testing for groups known to be at higher risk of TB. While this policy can be very effective, in practice it has many challenges. Most low-burden countries have abandoned the universal administration of BCG vaccines. Some countries like the Netherlands target children with a parent from a high-incidence country (WHO-estimated TB incidence > 50 per 100,000 person-years) for BCG vaccination.

Maintaining optimal services for child and adolescent TB in low-endemic settings requires persistent attention and advocacy. We provide an update on approaches to the detection and treatment of TB disease and infection in children and adolescents, from the perspective of a number of low-endemic settings. We highlight the ongoing challenges and opportunities to improve detection, treatment, and prevention by strengthening services, including those for special at-risk groups.

## 2. Epidemiology in Low-Burden Settings

There are many low TB-incidence countries located in various regions of the globe where exposure to *Mycobacterium tuberculosis* is now uncommon, and TB has not been a major cause of child morbidity or mortality for decades. Such countries share common challenges for maintaining TB services and, generally, still have vulnerable populations at higher risk of TB than the general population. Specific examples from our practice in North America and Western Europe are used to illustrate the epidemiology.

### 2.1. The United States

The overall incidence of TB in the United States declined slowly between the 1950s and early 1980s, when it increased (about 14% overall, but 20% in children and young adolescents) due to co-infection with HIV and diminished public health resources to conduct case-finding and contact tracing. A decline was reestablished in the mid-1990s and has persisted, but the rate of decline leveled off in the last few years. In 2019, there were 8920 cases at all ages [4]. The most comprehensive recent study of child and adolescent (0 to 17 years of age) TB evaluated all reported cases from 2007 to 2017 [5]. During this time, 68% of 5175 pediatric cases occurred among US-born persons in the 50 states, including 78% of the children and 42% of the adolescents; many of the US-born cases were the children of parents who were born in high TB-incidence countries and who recently migrated to the US. The overall annual rate was about 1 case per 100,000 of the child and adolescent population. Less than 1% of cases were known to be co-infected with HIV, and MDR-TB was reported in only 25 cases. However, culture confirmation was reported for only 69% of adolescents and 39% of children diagnosed with pulmonary TB. Overall, 68% of cases were pulmonary only, 22% were extrapulmonary only, and 9% were both. Contact tracing identified ~45% of the cases who were under 5 years of age, but only 28% of the cases who were 5 to 14 years of age. Only 14 deaths due to TB were reported. Approximately 66% of children who were under 15 years old and with TB would have been recommended for TB testing under the current targeted testing guidelines: 38% because of known recent contact, 21% for being born in a high-burden country, and 8% who had traveled extensively to a high-burden country. An additional 21% did not meet current guidelines for testing but had one or both parents who were born abroad.

In the same study, age-specific incidence rates were 12.9 times higher among non-US-born children and adolescents than among US-born children and adolescents [5]. Incidence rates also varied substantially among racial and ethnic groups. Rates among all other groups (Asian, Black, Hispanic, Native American or Alaskan, Native Hawaiian or Pacific Islander) were at least 14 times higher than among non-Hispanic white children and adolescents.

### 2.2. Canada

Unlike the declines seen in the United States, rates of childhood TB in Canada have remained stable over the past 7 years for which there are published data [6]. While childhood TB is largely a disease of the child who is either born or with parents born in a TB-endemic country in the US, and 66% of TB in all ages in Canada occurs in individuals not born in Canada, more than half of childhood TB in Canada occurs among Canadian Indigenous children, reflecting the very high rates and ongoing transmission in those communities [7]. The overall annual rate of childhood TB in Canada between 2013 and 2016 was 1.6/100,000 for all children, 10.2/100,000 for all Indigenous children (i.e., First Nations and Inuit and Metis children), and 100.8/100,000 among Inuit children [7]. In a national study, over 70% of childhood cases had a known contact, usually within the home [7]. Based on hospital and TB program data, childhood TB in the metropolitan immigrant-receiving areas is largely a disease of foreign-born children or the children of foreign-born parents [8]. The epidemiology of adolescent TB is difficult to determine as national reports include the entire 15–24-year range. Studies in Ontario province in 2002 suggested that adolescent TB is predominantly a disease of the foreign-born [9].

### 2.3. Western Europe

Most countries in the WHO European Region are low-burden countries with an all-age annual incidence of < 10/100,000 population [10]. In many of these countries, the large majority of cases can be attributed to the reactivation of TB infection acquired locally in the past or in high-endemic countries. The percentage of children with TB notified annually is correspondingly low. In 2019, 3.9% (range 0–7.1%, excluding one outlier country at 16.4%) of notified TB patients were children younger than 15 years of age, with a total of 1955 children or a TB rate of 4 per 100,000; 14 out of 29 reporting countries registered 20 or fewer children with TB, and 11 countries had fewer than 5 cases. Therefore, in most European countries, TB is a rare disease in children and many health care workers, including pediatricians, may not be confronted with TB in children during their professional careers. The high migrant influx has, however, importantly influenced TB epidemiology in many countries. For example, in Switzerland, annual TB incidence ranged from 1.4 to 2.8 per 100,000 child population between 2013 and 2019, which was comparable to earlier years [11]. In foreign-born children, incidence rates were considerably higher, up to 13.7 per 100,000, and peaked in 2016 [12].

Stringent contact investigation policies are in place in most countries. This is key to ensuring early diagnosis and preventive treatment in children. Not many countries report how many children with TB are detected through contact tracing. Available data, however, clearly show that TB cases detected through contact tracing are several-fold higher in children compared to adults. For example, in the Netherlands and in Germany, over 40% of the children diagnosed with TB were detected through contact tracing. In adults in Germany, this proportion was only 4% in 2019. For every child detected with TB disease in the Netherlands, 2.5 child contacts were started on TB-preventive treatment [13,14].

## 3. Current Approach to Case-Finding—TB Disease and Infection

### 3.1. Contact Tracing and Screening—Active Case-Finding

The data above demonstrate that in many low-burden countries, up to 50% of childhood TB cases are discovered via contact tracing or by screening specific risk groups [15]. In general, children with TB who are discovered this way have less extensive disease, very little extrapulmonary disease, and can be treated with the newer, shorter regimens. The disease may even be detected in subclinical stages, which recently have been reported more commonly [16]. Children can inhale *Mycobacterium tuberculosis* (Mtb) in a variety of settings, but in low-burden countries, it most often occurs in their own home or that of a close relative. The WHO End TB strategy has emphasized patient-centered care [17], but for prevention and early diagnosis of TB disease in children in low burden settings, it is important to emphasize family-centered care [1]. Contact tracing is a high-value activity: it identifies recently infected children, adolescents and adults who are at risk of rapid progression to TB disease; it detects early disease that is easier to cure; and it is the only opportunity to determine the likely drug susceptibility profile of the organism causing TB disease in children from whom the organism cannot be isolated, and infection in all child contacts [18,19]. The latter is particularly crucial when the source case has drug-resistant TB, and treatment regimens for the contacts with infection or disease need to be modified.

Unfortunately, the recent COVID-19 pandemic has significantly diminished public health resources in many low-burden settings, and TB programs have suffered some deleterious consequences [20,21]. As TB caseworkers are skilled in contact tracing, many were recruited early in the pandemic to perform contact tracing for COVID-19 patients. As a result, contact tracing for TB is more often delayed and incomplete. A recent survey of TB programs in the United States, conducted by the National Tuberculosis Controllers Association, revealed that over half of the programs reduced clinic hours and appointments, the proportion of contacts that were treated, and the number of sputum samples that were collected and processed; 35% of programs reduced the proportion of patients receiving DOT, and 64% reduced the processing of immigrants with evidence of possible latent TB infection (LTBI) [22]. It is crucial that the TB programs be replenished and upgraded to avoid a resurgence of both disease and mortality from it, and this is particularly important to protect exposed children.

The two groups of children and adolescents most likely to have LTBI in low-burden settings are the contacts of recent cases and those relocated from a high-burden country. Although universal testing for TB infection was discontinued in low-burden settings many years ago [23], it is important that children at higher risk than the general population should be identified and tested. Recent data suggest that one in 10 to 20 children with a refugee status have LTBI [24]. The most difficult aspect of identifying and testing for risk is that the vulnerable children are often the least likely to have a setting in which to receive routine care. For example, in the United States, persons with refugee status are automatically eligible for Medicaid health insurance, while other immigrant groups are not, often being employed at jobs that do not offer health insurance as a benefit [25]. As a result, the out-of-pocket costs for testing and treatment are beyond their reach. Unfortunately, public-health TB programs are so poorly funded that they often do not offer TB testing outside of contact tracing. One recent study in Houston, Texas, demonstrated that schools can be excellent settings in which to educate adolescents and teachers about TB, identify students with risk factors for LTBI and motivate them to get tested and, if necessary, be treated [26]. The cost-effectiveness of TB screening is key, and one study from Europe estimated that LTBI screening was cost-effective if LTBI prevalence was 14%, with a progression rate of 5% [27].

### 3.2. Evaluation of Ill Children and Adolescents—Passive Case-Finding

In low-TB-burden settings, clinical suspicion for TB in children and adolescents is often low; delays in diagnosis are common, despite repeated visits by patients to health services [28]. Language barriers have been associated with the presentation delay of TB in Europe [29], likely reflecting the difficulties of patients or parents in accessing care. Failure to consider the epidemiologic history may contribute to diagnostic delay [30]. In contrast to those diagnosed via active case-finding, children presenting passively with TB because of symptoms are older and have more severe pulmonary disease [15]. Extrapulmonary disease is common in symptomatic older children and adolescents in low-burden settings [9,31]; physician perception of TB as a pulmonary disease may delay diagnosis. In addition, children with evidence of infection and abnormal radiographs may be asymptomatic, as has been reported in one-third and up to half of younger children in studies from Switzerland, Austria, and Belgium [12]. Furthermore, TB may mimic more common conditions, including inflammatory bowel disease [32] and oligoarticular juvenile arthritis [33]. The pattern of extrapulmonary disease may differ between countries. For example, in the United States, most central nervous system TB has occurred in children younger than 2 years of age [34], whereas in Canada, older children and adolescents are more often affected [7]. Distinguishing those with pulmonary TB from patients with more common conditions may be difficult: in a recent study, the duration of symptoms was found to differentiate symptomatic adolescents attending a pediatric emergency department with infectious TB from those with pneumonia [35], but given the rarity of TB, routine algorithms to detect TB were considered difficult to implement. Failure to obtain sputum further delayed diagnosis.

## 4. Challenges in the Diagnosis of TB Disease

### 4.1. Microbiologic Techniques

Microbiologic confirmation of TB may not be obtained or even sought in children in low-burden countries. Children often have paucibacillary disease and multiple studies have shown that, on average, only 25% of clinical cases are confirmed, and, even with the best available techniques, microbiological confirmation can be attained only in 50–60% of cases [13,15,36,37]. Microbiologic confirmation is even lower for tuberculous meningitis and certain other extrapulmonary forms of the disease. However, in those children diagnosed via contact tracing who have only one known source case, microbiological confirmation may not be sought, and the drug susceptibilities of the source case’s isolate are used as a proxy to direct therapy for the child. When cultures are performed, gastric aspirates are frequently used for young children with intrathoracic disease. Although they can be performed as outpatient procedures [38], admission to the hospital is usually practiced, with resultant increased costs to the health service and parents. Multiple specimens are needed to increase the yield for culture or nucleic acid amplification (NAAT) [39]. Sputum induction in children older than 5 years requires appropriate isolation facilities that may not be available; in younger children, induced sputum may be a more convenient alternative or addition to gastric aspirates [40], but adequately trained personnel may not be available for proficiency in this procedure. More recent approaches of a combination of induced sputum with a following gastric aspirate performed on one day have shown comparable sensitivity to serial sampling over several days. In low-burden settings, because TB is not suspected, biopsy specimens of affected organs or tissues may not always be submitted for mycobacterial culture or NAAT. In high-burden countries, there is increasing focus on NAATs to diagnose childhood TB, although validation for extrapulmonary specimens is lacking [41]. Stool specimens and nasopharyngeal aspirates are other potential sources for detecting Mtb, but their use in most low-burden countries is in its infancy. The testing of stool specimens with NAATs may have a sensitivity of 50% or more compared with microbiologic or composite reference standards. However, while NAATs for TB are widely available in Europe and the United States [42], laboratories still need to develop protocols to process stool specimens. As is true in many high-burden countries, TB meningitis is often not confirmed microbiologically and a combination of microbiologic and immune-based tests—along with advanced radiographic studies—improves diagnosis [43].

### 4.2. Tests of Infection

Presently, two types of tests are commercially available to test children for infection with Mtb: the tuberculin skin test (TST) and interferon-gamma release assay (IGRA) [44,45]. Both tests measure the cellular immune response to Mtb. This requires the person to mount an immune response for it to work properly, for which an interval of 8 weeks after inhalation of the organism may be required. This delay may be a concern in children recently in contact with an infectious TB patient, as well as for young infants with an immature immune system, both groups being more likely to rapidly develop TB disease after infection [46].

TST and IGRA have a supportive role in the diagnosis of TB disease, as neither type of test distinguishes between TB disease and LTBI, and both have a poor predictive value for the development of TB disease after infection. The tests may be falsely negative or indeterminate in persons with TB disease, particularly among infants and younger children, patients with severe disease (especially in the case of meningitis and disseminated/miliary disease), and immunocompromised or malnourished children who are at a greater risk for severe forms of disease and death if they develop TB [47,48]. While the underlying mechanisms remain only partially understood, it is likely that incomplete immune maturation plays a significant role [49].

On the other hand, a test for infection can yield a false-positive result that potentially exposes the child to unnecessary evaluation and treatment. IGRAs are associated with fewer false-positive results than TST [44]. False-positive TST results are most often attributable to previous BCG vaccination or cross-reaction with environmental mycobacteria. In BCG-vaccinated children, testing with TST may boost the immune response to future TST results, resulting in false-positive results. In these children, confirmation of the test result with an IGRA is recommended. Most false-positive IGRA results are due to low-level nonspecific reactivity. Some TB experts have suggested that it is time to abandon the TST in favor of the IGRAs [50], while others are more skeptical [51]. The initial hesitation to changing over was due to a lack of proven sensitivity in the youngest children. However, as more data have become available, the IGRAs have been shown to be superior in the diagnosis of TB infection in low-burden locales when a major contributor to the burden of LTBI is in BCG-vaccinated persons [52].

### 4.3. Imaging

As in high-burden countries, the chest radiograph is an essential part of the diagnosis of TB in children and adolescents in low-burden countries [53]. Unfortunately, the specificity of the radiographic findings is low, meaning that when the incidence of TB in the community is low, the positive predictive value of some findings commonly seen with TB is also low. Studies have demonstrated that inter-observer agreement in the interpretation of the chest radiograph can be variable, even among practitioners with TB experience [54]. Many other infections and conditions can cause hilar or mediastinal adenopathy, and pulmonary parenchymal cavities in adolescents are more often in the form of pyogenic lung abscesses than TB. Knowing that the child was recently exposed to someone with TB, or recently came from a high-burden country, increases the likelihood that these findings are related to TB. Machine/artificial intelligence systems are becoming popular in some places, but relevant experience with young children is limited, and there is no evidence that they perform better in low-burden settings than human interpretation [55]. These systems are most useful when mass radiographic screening is employed, but this is seldom done or necessary in low-burden settings.

Ultrasound has also proven useful for the diagnosis of TB. It is often used to detect and characterize pleural effusions. Recent ultrasound studies have shown that the presence of abdominal adenopathy or splenic lesions suggests that TB may be the cause of associated abnormal chest radiographic findings, especially in children living with HIV [56,57].

One advantage for clinicians in low-burden settings is the availability of advanced imaging techniques. A computerized tomography (CT) scan is much more sensitive than plain chest radiography and can more readily distinguish among the various tissues in the thorax, making it better able to discern the presence of adenopathy, atelectasis, and pleural lesions [58]. A CT scan is not recommended for asymptomatic children with a positive test of infection and a normal chest radiograph, as the finding of small lymph nodes should not define disease or alter the therapy. However, a chest CT can be helpful when the chest radiograph is abnormal but it is unclear if TB is the cause. CT can also help indicate when TB may be the cause of extrapulmonary disease, particularly with the osteoarticular and abdominal forms of the disease [59,60]. CT has been used more frequently in recent studies in low-burden settings, but its indication should be carefully evaluated as it affords exposure to radiation, the application of contrast, and may require sedation in young children.

Magnetic resonance imaging has a variety of potential uses but the main one is for TB meningitis [61,62]. It is the most sensitive radiographic modality for detecting several hallmarks of central nervous system TB: basilar enhancement, hydrocephalus, cranial nerve involvement, infarcts, and small tuberculomas, particularly in the midbrain and infratentorial regions.

## 5. Challenges in the Treatment of TB Disease

### 5.1. Regimens

Treatment of TB disease in children in low-burden countries generally follows the WHO recommendations, and the regimens for children are generally the same as for adults. The treatment regimen most often used for uncomplicated, fully drug-susceptible TB disease is 2-month intensive therapy with a combination of isoniazid, rifampicin, pyrazinamide, and ethambutol, followed by a 4-month continuation phase of isoniazid and rifampicin [63,64]. Because multidrug-resistant TB disease is quite rare in children in low-burden countries, virtually all the known information about its optimal treatment comes from high-burden countries, and clinicians in low-burden countries tend to use regimens recommended by the WHO or the Sentinel group [65,66]. For TB meningitis, longer treatment has been recommended, which usually extends the continuation phase to 7 to 10 months.

Recently, the WHO has announced new recommendations for the treatment of TB in children, following review of evidence from recent studies. The SHINE study showed that the continuation phase can be shortened from four to two months among children with non-severe forms of TB [67,68]. The study showed that the shorter 4-month regimen was non-inferior to the 6-month regimen. Because so much childhood TB is found via contact tracing in low-burden countries, and those children commonly have non-severe disease, it is anticipated that the 4-month regimen will be used extensively in low-burden countries once the data from the SHINE trial are published and more widely disseminated. The new WHO recommendations also include a 6-month intensive treatment with high doses of rifampicin and isoniazid, the same dose of pyrazinamide, and ethionamide or levofloxacin replacing ethambutol, as an alternative to a 9- to 12-month regimen in children with TB meningitis. Finally, the WHO also now recommends the use of bedaquiline and delamanid in all age groups, so that the use of injectable medications should only be required rarely.

In general, medication administration is a major point of concern in the successful treatment of children. TB medications were designed for use in adults; giving medications to young children involves cutting or crushing pills, opening up capsules and making suspensions for which the pharmacokinetics and pharmacodynamics are unknown. To improve treatment adherence, the TB Alliance developed palatable and soluble fixed-dose combination tablets of first-line drugs that are available through the Global Drug Facility and are now used in many high-burden countries [69]. Regrettably, these child-friendly drugs are not available in many low-burden countries. However, pediatric formulations of ethambutol, rifapentine, and some second-line drugs will soon be available. High fees and complicated in-country regulatory mechanisms in a small consumer market are the major barriers to registration of the drugs in EU countries and North America [70]. As long as the child-friendly formulations are not available, parents or caretakers may need individual advice and coaching on how to administer the medication and what to do when medication is refused or vomited up by the child.

### 5.2. Drug Toxicity

While children generally tolerate TB medications better than adults, adverse drug reactions remain a significant problem during TB treatment. In a national Canadian survey of children treated for TB disease, 8.5% of patients experienced an adverse drug reaction (ADR) and 3% were hospitalized for a serious ADR. Pyrazinamide has been the most commonly implicated drug, causing hepatitis, severe pruritis and joint discomfort caused by elevated serum uric acid levels [71]. Severe isoniazid hepatotoxicity is rare in children and adolescents but has been reported, especially with poor monitoring and ongoing administration in the face of symptoms [72]. Isoniazid accounted for 14% of liver transplants for drug toxicity in the US in 2007 [73]. Clinicians treating TB, as well as patients and their caregivers, need to be made aware of the need to discontinue medications immediately and then access health services if symptoms that may represent an ADR occur. Management of patients by clinicians and teams experienced in treating tuberculosis may help minimize morbidity from drug toxicity [74].

### 5.3. Directly Observed Therapy and Other Support Measures

Directly observed therapy (DOT) has been a staple of TB management since the early 1990s, when the WHO made it part of its official short-course strategy [75]. It has become the standard of care in most low-burden settings for the treatment of TB disease and is frequently employed in the treatment of high-risk TB infections, especially in recent-contact cases, very young children, and immunocompromised individuals. DOT has taken many forms but, traditionally, has been via in-person contact between the health care worker and patient, either in the home or the clinic. It should be part of a package of support measures for the patient and family that may include the provision of transportation, help with food procurement, and other health services. There is no doubt that this has been a powerful strategy that has increased cure rates while lowering the rates of secondary drug resistance that can be caused by poor adherence to treatment.

However, despite its success, DOT has been criticized in some quarters as being coercive, limiting the autonomy of the patient and family and intruding on their schedules. The almost ubiquitous availability of cell phones has led to the recent development of video directly observed therapy (vDOT), which allows patients and families to record the taking of the medications at a time and place of their choosing, and then send the recording to the TB program via a phone app [76]. This approach has a high satisfaction rating and is an excellent example of the use of technology to address concerns regarding both the TB program and the patient and family.

It is important to view DOT and vDOT as part of a package of services to help and support patients undergoing treatment for TB. Many families dealing with TB in low-burden countries have other important issues to deal with, such as: poverty; lack of access to health care, especially preventive services; food insecurity; housing insecurity; joblessness; and limited education. In addition, families who have immigrated often have language issues and come from cultures that have different health beliefs than in the culture of their new country. It is critical that these other issues are acknowledged and, hopefully, addressed to some degree to ensure the success of TB therapy. TB programs need to have cultural sensitivity, access to language translation as needed, and establish and maintain linkages with other programs and agencies that can help support their patients and their families.

### 5.4. Clinician Experience

In low-burden settings, clinicians, including those trained in infectious diseases or respirology, may not have treated TB. Lack of experience with TB may influence patient outcomes: clinician experience with managing TB was shown to influence the mortality of TB patients in Canada [74]. Wherever possible, clinicians and teams experienced in managing pediatric TB should be involved in the care of patients with TB disease, either directly or through consultation. In the United States, the Centers for Disease Control and Prevention has established several regional Centers of Excellence that provide expert advice to clinicians managing TB in children and adults. In Europe, a similar network of support and expertise is provided by PTBnet.

## 6. Challenges in the Treatment of TB Infection

While much of the world opted to use BCG vaccination to try to prevent TB disease in children, some affluent countries used the alternative strategy of finding infected children and treating them, to prevent progression to disease. There are several regimens to treat TB infection in children and adolescents that are safe, effective, and relatively inexpensive by the standards of affluent countries [44,77]. Rates of significant adverse reactions are low, usually < 2%; significant hepatic dysfunction is uncommon, and most children are monitored clinically. Monitoring the laboratory measures of liver function is reserved for children who have underlying liver disease, are taking other potentially hepatotoxic medications, or are obese and at risk of fatty liver disease.

The greatest challenge to the treatment of TB infection in children and adolescents is adherence to therapy. Until the last decade, treatment of TB infection in children and adolescents was mired in the long-term, self-administered regimens with isoniazid. While the WHO standard has been 6 months (180 doses) of daily isoniazid, the United States opted for a standard 9-month (270 daily doses) regimen because older studies estimated an approximately 20% increase in protection with the longer course. Unfortunately, many studies have demonstrated poor adherence to these long regimens, with completion rates usually in the 50% to 70% range [78,79]. A 4-month regimen (120 daily doses) of rifampin is now used more frequently than isoniazid in many locales. Several studies have demonstrated improved adherence and completion rates with rifampin—80% to 90%—when compared with isoniazid therapy [80,81]. Although there have been no studies examining the long-term effectiveness of rifampin, short-term data suggest it is as effective as isoniazid. Two other regimens that combine isoniazid with a rifamycin drug are gaining in popularity. Several countries in Europe have used 3 months (90 daily doses) of isoniazid and rifampin with excellent tolerance and success [82]. The recent development of rifapentine, a rifamycin with a long half-life, has led to the regimen of 12 once-weekly doses of isoniazid and rifapentine (at least 11 doses in 16 weeks is considered adequate therapy). This latter regimen cannot be used in children under 2 years of age because of a lack of pharmacokinetic data for rifapentine in this age group; it is a difficult regimen for children under 5 years of age because of the current lack of a pediatric formulation. However, several studies have demonstrated treatment completion rates of greater than 90% with this regimen, with an effectiveness that is at least as good as other regimens [83,84]. This regimen can be used with DOT, vDOT, and self-administration, depending on the circumstances and available resources.

## 7. Special Groups to Consider

### 7.1. Adolescents

Studies suggest that many adolescents do not present themselves for evaluation of their TB symptoms in a timely manner [85]. Moreover, compared to other age groups, adolescents tend to have worse adherence to treatment for both TB infection and disease. Late presentation and suboptimal treatment adherence not only lead to increased morbidity and mortality in individual patients but also exacerbate TB transmission in the community—especially given adolescents’ often extensive social networks, tendency to congregate in groups, and high likelihood of having infectious forms of pulmonary TB. Adolescents at risk for TB may have other issues, such as substance abuse and pregnancy, that provide many challenges for treating TB infection and disease. Because TB disease and treatment can disrupt school and vocational training, adolescents’ education and future earning potential may suffer. Finally, the stigmatization of TB patients and the isolation that can be required at the beginning of therapy (to prevent further transmission) may negatively affect adolescents’ interpersonal relationships and self-esteem [86]. Many adolescents experience mental health issues—diagnosed or not—that may both impact and be impacted by TB care. Given the large number of adolescents affected by TB and the subsequent threats to their well-being, it is critical for healthcare providers, community leaders, and policymakers to optimize adolescents’ engagement in care and to minimize the harm they experience from TB infection, disease, and treatment [31].

A common model for viewing adolescent care and support was envisioned by Ross [87], and includes 5 basic domains: (1) good health; (2) connectedness and contribution to society; (3) safety and a supportive environment; (4) learning, competence, education, skills, and employability; and (5) agency and resilience. Adolescents truly fall between children and adults, and it is vital to invest in their care and to provide resources and services essential to their health. Because they are often reluctant to seek care, they should be sought out in active case-finding. Adolescents have an increased risk of treatment adherence challenges, including their loss to follow-up from TB care, and because TB treatment may interfere with their education and other developmental tasks, adolescents should receive the shortest effective TB treatment regimens. Efforts should be made to increase adolescents’ access to TB services, such as by offering after-school and weekend clinic hours, minimizing clinic wait times, providing community-based or decentralized TB care for adolescents, and facilitating easy transfer between TB care sites when adolescents need to relocate, such as for school, work, or changing living situations. Rifamycins render hormone-based contraception less effective, so TB providers should counsel adolescents accordingly, and provide or help adolescents to access alternative contraception methods. To the greatest extent possible, TB services should actively identify the wider healthcare needs of adolescents with TB and integrate TB care with other health services, such as within comprehensive adolescent health clinics. In the absence of co-located services, TB services need to develop clear referral pathways for common health concerns, such as reproductive healthcare, prenatal care, HIV care, treatment of substance use disorders, immunization, and mental healthcare.

### 7.2. Children Who Are about to Be Immunocompromised

An increasing number of children in low TB-burden affluent countries are treated for various conditions with immunosuppressing or immunomodulating medications. Obvious examples are cancer chemotherapy and high-dose or prolonged corticosteroid therapy, but many biological response modifiers—usually monoclonal antibodies or cytokine inhibitors—are increasingly used to manage a variety of rheumatologic, gastrointestinal, and dermatologic conditions. These drugs may also increase the likelihood of preexisting or acquired TB infection progressing to TB disease [88,89]. It is strongly recommended that children and adolescents who are about to be treated with these drugs should be tested for LTBI before receiving them, but adherence to this recommendation has often been poor. In addition, prior immunosuppression is common before biological agents are used, subsequently leading to false-negative TST or IGRA results. In a multicenter retrospective pediatric tuberculosis European trials group (ptbnet) study, 19 cases of pediatric TB related to anti-tumor necrosis factor-alpha (anti-TNF-alpha) therapy were identified over 30 months [47]; most patients had severe disease, and in one case, TB was diagnosed postmortem. Fifteen of these patients had testing prior to anti-TNF-alpha therapy but only one had a positive test, and most were receiving some immunosuppressive therapy before testing. It is important that screening for LTBI should occur at the time the primary diagnosis is made and before immunosuppression is undertaken. The risk of a false-negative test due to immunosuppressive medication should be considered in children with a high epidemiological risk of TB exposure and especially in those with a history of TB contact.

The American Academy of Pediatrics recommends that testing for a child about to be treated with immunosuppressive therapy can be performed with either the TST or an IGRA, and testing should occur regardless of whether the child has other epidemiologic risk factors for LTBI [44]. Some experts recommend that a single test should be performed for children with no other TB risk factors, but that both the TST and an IGRA should be performed for children with additional TB risk factors to maximize sensitivity (although this will create more false-positive results). Any positive test that is performed for this purpose should be considered to represent LTBI, and the child should be evaluated and treated accordingly.

## 8. Education and Advocacy—Roles for Professional Organizations, Professional Schools, Non-Governmental Organizations

TB public health programs usually have very limited resources, and it is essential that clinicians advocate on their behalf. Economic studies have demonstrated that TB program activities in general—and especially contact tracing and the treatment of TB infection—are efficient and cost-saving. Unfortunately, as case numbers decline, there is a tendency to try to diminish the resources available to government-supported TB programs. However, maintenance of an adequate infrastructure is crucial to maintain success [90,91]. As with the elimination of smallpox, if the elimination of TB is the goal, the most expensive case will be the last one, and basing funding solely on cost-effectiveness will doom the effort to failure.

Other organizations have important roles in the drive to eliminate TB. One of the most important roles is the education of health care providers. Prior to the resurgence of TB, seen in association with the beginning of the HIV pandemic, medical and nursing schools had relegated the teaching of TB to a handful of lectures. In the mid-1980s and early 1990s, as the association between the risks of TB and HIV were seen as intertwined, TB education became much more robust. Faculty members engaged in more TB–related research as more funding was made available, and they brought this to their teaching. Unfortunately, as TB rates have again declined in low-burden countries, the amount of research, available funding, and educational content have also receded again. There has been a notable decline in the number of presentations related to TB at international meetings of the American Thoracic Society and Infectious Disease Society of America, reflecting diminished funding and interest. It is imperative that minimal standards of TB-related education are established and maintained by professional schools and organizations so that future clinicians will continue to seek out TB risk factors in their patients and evaluate them accordingly [90].

Many non-governmental organizations remain active in the fight against TB. In North America, STOP TB Canada and STOP TB USA help coordinate resources and provide professional education, while examples based in Europe include The Union and KNCV. The Treatment Action Group, RESULTS, and the TB Roundtable are among the organizations that advocate for increased public funding for TB services, including those for children and adolescents. Finally, We Are TB is made up of TB survivors and the parents of affected children, who educate the public and elected officials about the need to bolster TB services.

## 9. Research Considerations

It is difficult to conduct randomized clinical trials pertaining to TB disease in low-burden settings due to the paucity of cases. Retrospective and observational studies have supplied some useful information, especially about the implementation of recommended regimens. Canadian [7] and regional collaborations, such as the Pediatric Tuberculosis European Trials Group [47], have produced representative studies with adequate sample sizes that deal with the unique problems of these settings; it is imperative that these and similar efforts be supported and funded. The strengths of low-burden settings that may uniquely contribute to TB research include more complete and longer follow-up, the ability to closely monitor for drug toxicity and to accurately determine the causes of morbidity and death. Trials of treatment for TB infection have been conducted because of the greater volume of subjects that are available and the presence of an organized structure in which the studies can be conducted. Much of the needed research is operational in nature, determining the optimal way to utilize available tools. Given the low incidence of TB disease in general, and TB infection in most groups, paying proper attention to the specificity of new tests is critical to avoid large numbers of false-positive results, which also can be minimized by restricting testing for LTBI to individuals with recognized risk factors.

## 10. Conclusions

Although the nature of TB disease may be similar in children in various settings, most general pediatricians and many subspecialists practicing in low-burden countries will never see a case. Enhanced educational efforts and guidelines are needed to encourage clinicians to think about TB infection and disease in their pediatric patients. Experienced clinicians and teams should, wherever possible, be involved in the management of childhood TB disease. Because microbiological confirmation of the disease is so difficult, the diagnosis is often delayed while other, more common infections and conditions are considered and excluded. Clinical detection of the symptomatic child with TB is poor in all settings; childhood TB disease is best prevented. In low-burden settings, contact tracing is of paramount importance and, when conducted well, prevents both mild and damaging disease. Contact tracing discovers a large proportion of cases, the cases tend to be non-severe and can be treated with shorter regimens, and it finds recently infected children who are at the highest risk of developing TB disease. Thus, strengthening TB services for children and adolescents requires preserving and protecting the primal importance of the public health infrastructure. Cutbacks in public health TB funding because of declining rates of TB have historically, and are again, likely to lead to a resurgence of disease and reverse the gains made in some low-burden countries. Finally, since most pediatric TB in low-burden countries is related to immigration, it is in the immediate and long-term interest of low-burden settings to contribute to the global struggle to end TB in high-burden countries [92].

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
