# Peer review of "Strengthening Tuberculosis Services for Children and Adolescents in Low Endemic Settings"

_pathogens, 2022, doi:10.3390/pathogens11020158_

Round 1

Reviewer 1 Report

Thank you for the opportunity to read this important article, entitled “Strengthening tuberculosis services for children and adolescents in low endemic settings.” This is an incredibly important topic and I am excited to see it addressed so clearly by the authors. The authors do an excellent job at identifying existing gaps within the health systems of low-burden settings that would benefit from strengthening to better serve children and adolescents at risk of TB infection and disease. All evidence is laid out clearly, is well-cited, and is thorough. Additionally, the manuscript is well-written and flows nicely. Kudos to the authors for a truly superb article.

Author Response

Thank you for your kind comments!

Reviewer 2 Report

This is a detailed and wide-ranging review of the considerations for treating children and adolescents with TB in low-burden settings.  The article does a good job of highlighting management considerations specific to low-burden settings. The only section in which I am recommending major revision is Section 2 (epidemiology); I find this section problematic for a couple related reasons, described in major comments 1-4 below.  Otherwise, I think the article is very readable and offers insight into the challenges of maintaining capacity to diagnose and manage a serious illness that is becoming progressively rarer.

Major comments:

1) No rationale for the choice of countries discussed (US, Canada, selected countries of western Europe) is given. The rationale does not seem to be estimated TB incidence. Several Middle Eastern countries (Israel, Jordan, Oman, UAE) have TB incidences <10 per 100k (the threshold mentioned for considering western European countries to be low burden).  Australia and New Zealand also fall below this threshold, with estimated incidences lower than several western European countries.  Costa Rica and Japan are very close to this threshold as well, with estimated TB incidences lower than Portugal.  Without an explanation, the discussion of only the US, Canada, and western Europe seems to support a white ethnocentric and colonial lens, which has historically been a problem within the TB community; why not acknowledge that there are countries in multiple regions of the world where TB is no longer a major public health threat? 

2) Because this section comprises a detailed discussion of the epidemiology of a non-representative set of example countries, and different information is presented in each sub-section, the take-home message is about TB epidemiology in low-burden settings is not clear. I would suggests revising Section 2 to focus not on geographic areas but on general epidemiologic themes that are common across low-burden settings, providing examples from specific countries to illustrate.  This would also help address comment 1, as a larger range of countries could be incorporated into the discussion without increasing the length of the article.  To me, some common themes that emerge in the existing section are that (1) TB incidence is low among children (generally lower than among adults?), (2) TB incidence is higher among people who have moved to low-burden countries from high-burden countries and their families (including children born in the low-burden country), (3) racial disparities exist, some of which is associated with immigration, and indigenous communities are also at particular risk (this is mentioned for Canada, but the same is true for the US, Australia, and New Zealand) and (4) a large proportion of children who are diagnosed are found through contact tracing. 

3) The sentence starting at line 79 says “While childhood TB is largely a disease of the foreign-born or the children of foreign-born parents in the US…” but the sentence starting at line 58 says that 68% of pediatric cases were among US-born individuals. These statements seem contradictory, although I suspect that they are not. I assume that the issue is that the first sentence considers US-born children of immigrant parents in the same group as immigrants while the second sentence distinguishes only by country of birth. But reading these two sentences in succession makes the take-home point unclear: is TB among children and adolescents in the US largely confined to immigrant communities? Or do the authors want to make the point that TB in children is *not* confined to people born in other countries? This comment may be irrelevant if the authors revise the section in response to comment 2, but I am including it in case the authors choose not to change the focus on their chosen geographic areas.

4) I understand that “foreign-born” is a standard term used in low-burden countries to describe anyone not born that country, and that it is a useful binary variable for surveillance systems.  However, in my experience working in the US, I find that the term often comes with racial and stigmatizing undertones, and some TB programs have decided to stop using it in favor of “individuals not born in the US” or the more specific “individuals born in high-burden countries.”  I would encourage the authors to reconsider the use of this term, which predominantly occurs in Section 2.

5) The third paragraph of the TST/IGRA section has no citations for the factual statements in lines 238-246. Many of these statements are things for which people often look for documentation – I think it could be especially useful for people working in higher-burden settings who are exploring the pros and cons of switching to IGRA from TST.   I suggest adding citations for at least some of the statements such as TST cross reactivity with BCG or environmental mycobacteria, boosting effect of TST, and non-specific reactivity as a cause of false-positive IGRAs.

6) I do not understand the sentence at lines 240-242 “In non-BCG vaccinated children, the TST has still its place, as IGRAs may be falsely negative in a small proportion of infected children.”  This only makes sense if the IGRA may be falsely negative in a specific subset of children in whom the TST would be positive, but it is not clear to me from the rest of the paragraph who this group of children is or why the TST would be positive while the IGRA is negative. 

Minor comments

  • The section headers are generally sentence-case (capitalizing only the first word). However, in the section 7 header, the word “Groups” is capitalized and in the section 9 header the word “Considerations” is capitalized; should these be lower-case?
  • At line 302, the “Sentinel Group” is referred to, but the author of the corresponding citation is the “Sentinel Project for Pediatric Drug-Resistant Tuberculosis.” To avoid confusion about the name of the group, perhaps “group” should be lower-case?
  • At line 428, I think the word “TB” may be missing after “infectious forms of pulmonary”

Author Response

Thanks for review.

Point-by-point response in attached file.

Reviewer 3 Report

This is a well written paper; I have only some minor observations: • Please standardize number format and separate thousands with spaces or commas, but not both. In example: Line 85: “10.2/100,000”, Line “100.8 /100 000”. • Line 94: Please define what “EU” and “EEA” means. • Finally, please check references according to the journal requirements.

Author Response

attached response.
